

# Convergence of hydrodynamic modes: insights from kinetic theory and holography

**Michal P. Heller[1,2⋆], Alexandre Serantes[2†], Michał Spaliński[2,3‡], Viktor Svensson[1,2∘] and Benjamin Withers[4§]**

**1** Max Planck Institute for Gravitational Physics (Albert Einstein Institute),
14476 Potsdam-Golm, Germany
**2** National Centre for Nuclear Research, 02-093 Warsaw, Poland
**3** Physics Department, University of Białystok, 15-245 Białystok, Poland
**4** Mathematical Sciences and STAG Research Centre, University of Southampton,
Highfield, Southampton SO17 1BJ, UK

⋆ michal.p.heller@aei.mpg.de, † alexandre.serantesrubianes@ncbj.gov.pl,
‡ michal.spalinski@ncbj.gov.pl, ∘ viktor.svensson@aei.mpg.de,
§ b.s.withers@soton.ac.uk

## Abstract

We study the mechanisms setting the radius of convergence of hydrodynamic dispersion relations in kinetic theory in the relaxation time approximation. This introduces a qualitatively new feature with respect to holography: a nonhydrodynamic sector represented by a branch cut in the retarded Green's function. In contrast with existing holographic examples, we find that the radius of convergence in the shear channel is set by a collision of the hydrodynamic pole with a branch point. In the sound channel it is set by a pole-pole collision on a non-principal sheet of the Green's function. More generally, we examine the consequences of the Implicit Function Theorem in hydrodynamics and give a prescription to determine a set of points that necessarily includes all complex singularities of the dispersion relation. This may be used as a practical tool to assist in determining the radius of convergence of hydrodynamic dispersion relations.

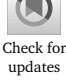

# 1  Introduction

Understanding the foundations of relativistic hydrodynamics as a description of nonequilibrium physics has been an important research theme of the past decade. The experimental motivation behind it comes from the field of ultrarelativistic heavy-ion collisions, where relativistic hydrodynamics is the framework successfully used to connect the early time physics of quantum chromodynamics (QCD) with the properties of the particle spectrum in the detectors [1,2]. On the theoretical side, how the hydrodynamic regime emerges from QCD in particular and quantum field theories in general has turned out to be a subject ripe for discoveries.

Our work is motivated by three interrelated lines of theoretical physics research. The first one concerns microscopically accurate descriptions of strongly-coupled quantum field theories with large number of degrees of freedom in terms of their classical gravity duals [3]. The second one concerns partial differential equations that embed relativistic hydrodynamics in a form of well-behaved and numerically tractable equations of motion for relativistic matter. We will refer to such frameworks as MIS-type models [4]. The last one concerns relativistic kinetic theory, which may arise as an effective description of QCD, or as a standalone model. The common feature among them is the absence of stochastic effects.

Broadly speaking, there are two main strategies to explore the transition to hydrodynamics in these different frameworks. The first involves studying highly symmetric flows, with the boost-invariant (Bjorken) flow being a widely used model of expanding matter in ultrarelativistic heavy-ion collisions [5]. The second utilizes linear response theory studies of singularities of retarded two-point functions $G_R$ in the complex frequency plane at a fixed spatial momentum. In our work we will be mostly concerned with the latter situation and we will return to expanding plasma systems only in the summary.

The key observation of holography is that in a class of quantum field theories, the retarded two-point functions of the stress tensor in equilibrium have singularities in the form of infinitely many single poles [6]. Each pole gives rise to a decaying and oscillating contribution upon inverting the Fourier transform. A similar story holds in MIS-type models, in which the number of such singularities is finite and small. As a result, some features encountered in holography can also be understood in these settings where they are often analytically tractable.

Among such poles, there are two special ones, which are arbitrarily long-lived upon making spatial momentum sufficiently small. These are the hydrodynamic shear and sound modes characterized by gapless dispersion relations of the respective form,

$$\omega_\perp = -i\frac{\eta}{sT}k^2 + \mathcal{O}(k^4) \quad \text{and} \quad \omega_\parallel^\pm = \pm\frac{1}{\sqrt{3}}k - i\frac{2}{3}\frac{\eta}{sT}k^2 + \mathcal{O}(k^3), \tag{1}$$

where $\eta/s$ is the ratio of shear viscosity to entropy density and $T$ is the equilibrium temperature. The small-$k$ expansion is a direct momentum space manifestation of the hydrodynamic

gradient expansion and in (1) we dropped contributions from terms having two and more derivatives of fluid variables. The other modes are exponentially decaying in time and correspond to transient effects when perturbing equilibrium by a small amount. For holographic models, hydrodynamic and transient excitations are nothing else than quasinormal modes of anti-de Sitter black holes with planar horizons [7].

In the context of the aforementioned foundational aspects, the question that rose to prominence in the past two years, see [8–16], is what is the radius of convergence of the hydrodynamic dispersion relations when expanded around $k = 0$ and, if finite, what sets it.

It turns out that in the holographic quantum field theories studied to date starting with [8], as well as in the MIS-type models analyzed so far [12], the radius of convergence, $|k^*|$, is finite and the critical momentum it corresponds to, $k^*$, is set by a branch point singularity of the hydrodynamic dispersion relation. This branch point is the result of a collision between the hydrodynamic mode in question and one of the transient modes in a complexified spatial momentum plane. By this we mean that the single pole singularities of the retarded correlator move in the complex frequency plane as a function of momentum and for some momenta they degenerate (collide).

Note that in linear response theory it is sufficient to study only the radius of convergence of the small-$k$ expansion of the dispersion relations. This is because this radius also dictates the radius of convergence of the position-space gradient expansion of the hydrodynamic constitutive relations as shown in [12], when supplemented with a choice of initial data.

Holography is a framework dealing with strongly-coupled quantum field theories and a natural question that arises is how the story of modes and their collisions generalizes to weakly-coupled situations. This is addressed in our present work. The starting point for our considerations is relativistic kinetic theory

$$p^\mu \partial_\mu f = \mathcal{C}[f], \tag{2}$$

where $f$ is a one-particle distribution function, that depends on spacetime coordinates and the particle momenta $p^\mu$, and $\mathcal{C}$ is a collision kernel that defines the model and encodes interactions between particles.[1]

Kinetic theory can act as an effective description of processes in weakly-coupled quantum field theories when interference effects can be neglected [17, 18]. In particular, the effective kinetic theory of QCD [19–21] has recently become a framework of significant theoretical and phenomenological interest in the context of ultrarelativistic heavy-ion collisions [22–31].

In general, little is known about the inner workings of retarded correlators in kinetic theories with nontrivial collision terms (see, however, [32]). In the present work we will specialize to a particularly simple and, therefore, widely employed collision kernel that encodes exponential relaxation to the equilibrium distribution function

$$\mathcal{C}[f] = p_\mu u^\mu \frac{f(x,p) - f_0(x,p)}{\tau}, \tag{3}$$

where $u^\mu$ is the comoving velocity vector, $\tau$ is the relaxation time and we take $f_0(x,p)$ to be given by a Boltzmann distribution

$$f_0(x,p) = \frac{1}{(2\pi)^3} e^{\frac{p_\mu u^\mu}{T}}. \tag{4}$$

For this so-called relaxation time approximation (RTA) kinetic theory [33] and for massless particles, the retarded correlators were computed in closed-form in [34] (see also [35]). If

---

[1]Throughout the text we assume mostly plus metric sign convention.

$x^0$ denotes time and $k$ is the Fourier transform momentum component along the $x^3$ direction, then the retarded correlator in the shear channel takes the form

$$\frac{G_{R,\perp}^{01,01}(\omega,k)}{-(\mathcal{E}+\mathcal{P})} = \frac{2k\tau(2k^2\tau^2 + 3(1-i\tau\omega)^2) + 3i(1-i\tau\omega)(k^2\tau^2 + (1-i\tau\omega)^2)L}{2k\tau(3 + 2k^2\tau^2 - 3i\tau\omega) + 3i(k^2\tau^2 + (1-i\tau\omega)^2)L},$$
(5)

whereas in the sound channel one gets

$$\frac{G_{R,\parallel}^{03,03}(\omega,k)}{-3(\mathcal{E}+\mathcal{P})} = \frac{1}{3} + \omega^2\tau\frac{2k\tau + i(1-i\tau\omega)L}{2k\tau(k^2\tau + 3i\omega) + i(k^2\tau + 3\omega(i+\tau\omega))L},$$
(6)

with $L$ denoting the logarithmic term

$$L = \log\left(\frac{\omega - k + \frac{i}{\tau}}{\omega + k + \frac{i}{\tau}}\right).$$
(7)

In the above equations $\mathcal{E}$ and $\mathcal{P}$ are equilibrium energy density and pressure. The remaining nontrivial components can be obtained using tracelessness and conservation of the stress tensor. In the rest of the text, if not explicitly stated, we will set $\tau = 1$ without loss of generality.

While, unsurprisingly, one finds that there exist shear and sound mode frequencies arising as single pole singularities of respectively (5) and (6), the correlators also contain logarithmic branch point non-analyticities. The corresponding branch cuts emanate from $\omega = -\frac{i}{\tau} \pm k$ and, therefore, represent a transient sector whose real-time imprint is an exponential decay supplemented with oscillations and power-law corrections [4].

Following [35], one can understand the branch cuts as originating from the free propagation of particles whose interactions with the background equilibrated medium are captured by their finite lifetime set by $\tau$. The branch cut arises because perturbations of the stress tensor at a given spatial point receive contributions from particles moving with the speed of light and coming in from various directions. The latter are single pole contributions that the integration over angles converts to the logarithm (7). Because one deals with a correlator of a conserved quantity, the contribution of the decaying particles to the stress-energy tensor cannot be lost and needs to be transferred to other degrees of freedom – the hydrodynamic shear and sound waves.

The aim of our study of kinetic theory is to understand the interplay between the hydrodynamic modes – described by single poles which, at low $k$, are localized close to the origin in the complex $\omega$-plane – and the branch cut transient sector (7). In particular, we want to understand what kind of phenomena set the radius of convergence for the hydrodynamic dispersion relations in this setup. In [34, 35] it was noticed that since the correlator (5)-(6) contains a branch cut, the hydrodynamic poles can move to a non-principal sheet as a function of (in these works, real) momentum, labelled as the 'hydrodynamic onset transition'. However, since the position of branch cuts is largely a matter of choice, we do not anticipate that this transition is related to the radius of convergence of the hydrodynamic expansion, and indeed it is not.

Starting from the explicit expressions of the retarded thermal two-point functions contained in equations (5)-(6), we will provide substantial numerical evidence supporting the fact that, in RTA kinetic theory, the mechanism determining the critical momentum $k^*$ is different from holography. Our main findings are the following:

1. In the shear channel, the radius $|k_\perp^*|$ is set by a collision between the hydrodynamic pole $\omega_\perp(k)$ and a nonhydrodynamic branch point $\omega_{bp}(k)$. This corresponds to a logarithmic branch point singularity of $\omega_\perp(k)$. We find $|k_\perp^*| = 3/(2\tau)$.

2. In the sound channel, the radius $|k_\parallel^*|$ corresponds to a collision between the hydrodynamic pole $\omega_\parallel^\pm(k)$ and another *gapless* pole that originates in a non-principal sheet of

the retarded correlator. The collision itself takes place on a non-principal sheet of the correlator, and corresponds to a branch point singularity of $\omega_\parallel^\pm(k)$. We find numerically $|k_\parallel^*| \simeq 0.7410387/\tau$.

With the mechanisms in both RTA kinetic theory and holography established, we turn our attention to general lessons. We use the Implicit Function Theorem to place constraints on the singularities of the dispersion relation $\omega(k)$ in any theory where they can be defined by an equation of the form $P(\omega, k) = 0$. Both the location of the singularities and the type of singularity can be constrained by the form of $P$.

The structure of the paper is as follows. The shear and the sound channel dispersion relations in RTA kinetic theory are respectively discussed in sections 2 and 3. Afterward, in section 4 we comment on our results in light of the Implicit Function Theorem, where we also provide a general prescription for determining the radius of convergence of the hydrodynamic modes. We close the paper with a summary in section 5.

## 2   The shear channel

In the shear channel, the poles of the retarded thermal two-point function (5) correspond to the solutions of

$$P_\perp(\omega, k) = 2k(3 + 2k^2 - 3i\omega) + 3i(k^2 + (1 - i\omega)^2)L = 0. \tag{8}$$

For future reference, we define $A(\omega, k) = 2k(3 + 2k^2 - 3i\omega)$ and $B(\omega, k) = 3i(k^2 + (1 - i\omega)^2)$. Apart from the hydrodynamic mode $\omega_\perp(k)$, that behaves as

$$\omega_\perp(k) = -\frac{i}{5}k^2 + \dots , \tag{9}$$

when $k \to 0$, (5) is endowed with two nonhydrodynamic branch points located at

$$\omega_{bp}^\pm(k) = \pm k - i. \tag{10}$$

Our main focus is the large-order behavior of the series expansion (9). Introducing the ansatz

$$\omega_\perp(k) = \sum_{q=1}^\infty c_q k^{2q} \tag{11}$$

into (8), expanding around $k = 0$, and demanding that the resulting series vanishes order-by-order, we can find the $c_q$ coefficients straightforwardly. We have carried out this procedure up to $q = q_{max} = 500$. The results of applying the ratio test to the sequence $\{c_q, q \in \mathbb{N}\}$ are plotted in figure 1 (left). We find that the norm of the critical momentum in the shear channel, which must satisfy

$$\lim_{q \to \infty} \left| \frac{c_{q+1}}{c_q} \right| = |k_\perp^*|^{-2}, \tag{12}$$

is compatible with the value

$$|k_\perp^*| = \frac{3}{2}. \tag{13}$$

This is illustrated in the left plot in figure 1 by a dashed blue line corresponding to $|k_\perp^*|^{-2} = \frac{4}{9}$, as well as in the right plot, where we show that the difference between $\left| \frac{c_{q+1}}{c_q} \right|$ and $|k_\perp^*|^{-2}$ indeed goes to zero as $q \to \infty$. In order to cross-check this result, we analytically continue the series expansion (11) into the complex $k$-plane by means of symmetric Padé approximants,

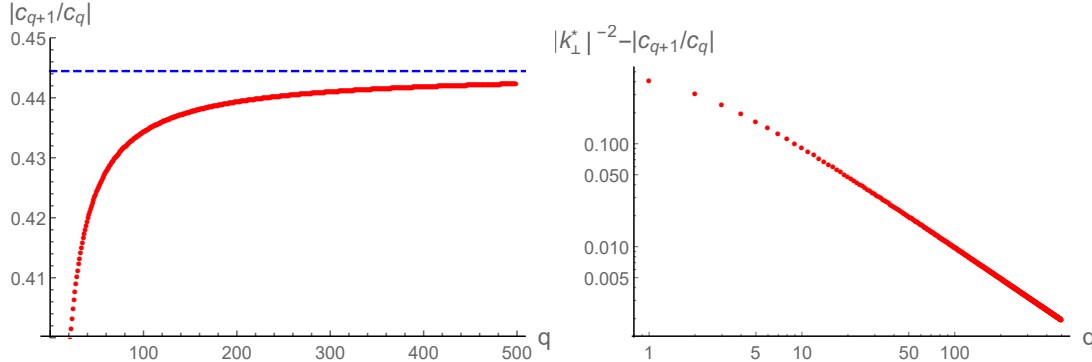

Figure 1: Left: ratio test applied to the sequence $\{c_q, q \in \mathbb{N}\}$. As $q$ increases, the quantity $\left|\frac{c_{q+1}}{c_q}\right|$ approaches a constant. The conjectured limit as $q \to \infty$, $|k_\perp^*|^{-2} = 4/9$, is represented by the dashed blue line in the plot. Right: difference between $|k_\perp^*|^{-2}$ and $\left|\frac{c_{q+1}}{c_q}\right|$ as $q \to \infty$. We clearly see that this difference tends to zero as a power-law.

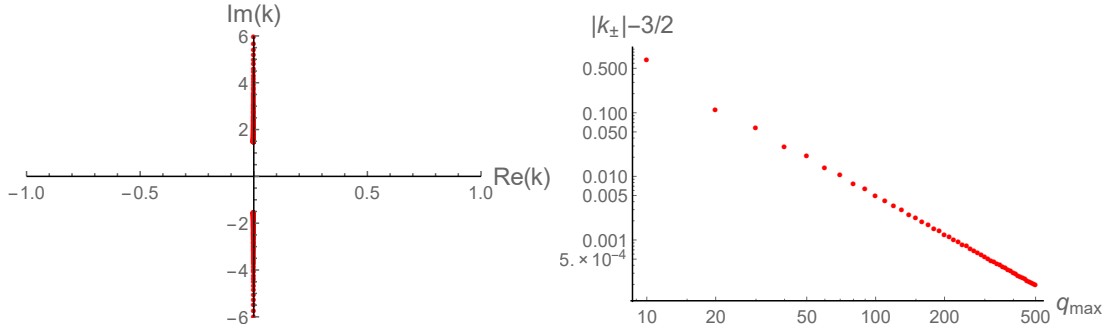

Figure 2: Left: location of the poles of the symmetric Padé approximant of order 250 in the complex $k$-plane. Two lines of pole condensation along the imaginary $k$-axis are clearly visible. For the resolution employed here, these lines start at $k = \pm 1.50020004i$. These values are compatible with $|k_\perp^*| = \frac{3}{2}$. Right: difference between $|k_\pm|$ and $|k_\perp^*|$ as the order of the symmetric Padé approximant, $\frac{q_{max}}{2}$, is increased.

and determine the single poles of the resulting rational function. In this approach, a branch point singularity of the exact $\omega_\perp(k)$ manifests itself as a line of pole condensation. The results of this procedure, for a symmetric Padé approximant of order 250, are shown in figure 2 (left). We find two lines of pole condensation along the imaginary $k$-axis, starting at

$$k = k_\pm = \pm 1.50020004i, \tag{14}$$

in very good agreement with the observations performed above. Furthermore, upon increasing the order of the symmetric Padé approximant, the difference between $|k_\pm|$ and $|k_\perp^*| = \frac{3}{2}$ decreases monotonically in a power-law fashion, as can be seen in figure 2 (right).

In the light of the results presented so far, it is natural to conjecture that the series expansion of $\omega_\perp(k)$ around $k = 0$ stops converging due to the presence of two branch point singularities located at $k = \pm\frac{3}{2}i$.

It can be checked directly that (8) vanishes at $k = \pm\frac{3}{2}i$, $\omega = \frac{i}{2}$, since the divergence of the logarithmic term as $\omega \to \frac{i}{2}$ is suppressed by its $\omega - \frac{i}{2}$ prefactor. Hence, the points $\left(k = \pm\frac{3}{2}i, \omega = \frac{i}{2}\right)$ are valid solutions of $P_\perp = 0$. These points are special from several viewpoints. First, when $k = \pm\frac{3}{2}i$, the nonhydrodynamic branch point $\omega_{bp}^\pm$ is located at $\frac{i}{2}$, which

implies that at $k = \pm\frac{3}{2}i$ the hydrodynamic pole collides with a nonhydrodynamic branch point, as claimed in the Introduction. Second, since these points can also be found by demanding that

$$A(\omega, k) = B(\omega, k) = 0, \tag{15}$$

at nonzero $k$, they also correspond to the only finite momentum solutions at which the coefficient of the logarithmic branch point present in (8) disappears.

Let us illustrate the first point mention above. We will do this by computing numerically $\omega_\perp$ along the ray $k = k_\theta \equiv |k|e^{i\theta}$, calculating its distance to the $\omega_{bp}^+$ branch point, and showing that this distance vanishes as $\theta \to \frac{\pi}{2}$.

Before presenting our results, let us emphasize an important observation that needs to be taken into account in order to carry out this computation. As originally found in reference [34] for the real $k$ case, $\omega_\perp$ can cross the branch cut joining $\omega_{bp}^+$ and $\omega_{bp}^-$.[2] This crossing does not entail that $\omega_\perp$ ceases to exist [35]; it just means that $\omega_\perp$ migrates to a different sheet of the retarded two-point function defined by analytical continuation. While this branch cut crossing does not pose any obstruction to the convergence of the series expansion of $\omega_\perp$ around $k = 0$, it has to be taken into account when obtaining $\omega_\perp$ numerically.

There are essentially two different ways to achieve this. The first one is to trade (8) for an ODE for $\omega_\perp$; the second, to analytically continue $L$ to a non-principal branch once the branch cut crossing has taken place. In particular, to go the $n$-th sheet of $P_\perp$, one just has to replace $L$ as given in (7) by

$$L_n = \log\left(\frac{\omega - k + i}{\omega + k + i}\right) + 2\pi i\, n. \tag{16}$$

We have verified explicitly that both approaches are compatible with one another.

To get the ODE, we calculate

$$\frac{d}{dk}P_\perp(\omega_\perp(k), k) = \partial_\omega P_\perp(\omega_\perp(k), k)\omega_\perp'(k) + \partial_k P_\perp(\omega_\perp(k), k) = 0, \tag{17}$$

and employ (8) to replace the logarithmic term in (17).[3] The final result is that

$$k(i - 2\omega_\perp)\omega_\perp' + 3\omega_\perp(i + \omega_\perp) - k^2 = 0. \tag{18}$$

This equation is to be solved with initial conditions given by the hydrodynamic shear mode small-$k$ expansion. Some results for $|\omega_\perp(k_\theta) - \omega_{bp}^+(k_\theta)|$ when $\theta = \frac{\pi}{2} - \delta\theta$, $\delta\theta > 0$ are shown in figure 3. Our findings confirm our expectations: in the limit $\delta\theta \to 0$, the hydrodynamic pole and the branch point collide at $|k| = \frac{3}{2}$.[4] An equivalent plot can be obtained by monitoring the distance between $\omega_\perp$ and $\omega_{bp}^-$ along the ray defined by $\theta = -\frac{\pi}{2} + \delta\theta$.

We would also like to offer an alternative way of picturing the branch point, hydrodynamic pole collision, more in line with the observations around equation (15). This alternative picture builds upon the crucial fact that the structure of the solutions of (8) can change qualitatively if we analytically continue $P_\perp$ to other sheets. For instance, gapless solutions require $n = 0$ and hence do not exist in the non-principal sheets; at the same time, while gapped solutions are absent in the principal sheet, they do appear when $n \neq 0$.

Let us focus on these gapped solutions. By performing the replacement $L \to L_n$ in $P_\perp$ (as mentioned in equation (16)) it is possible to obtain them as the following series expansion around $k = 0$,

$$\omega_\perp^{(NH,n)}(k) = -i + \alpha k + \frac{1}{3}i(\alpha^2 - 1)k^2 + \frac{1}{81}i(\alpha^2 - 1)^2 k^4 + \frac{1}{243}\alpha(\alpha^2 - 1)^2 k^5 + \dots, \tag{19}$$

---

[2]Unless stated otherwise, we will always consider that the log function is evaluated on the principal branch.

[3]The sequence of steps involved in obtaining the equation breaks down right at the point where the hydrodynamic mode collides with the branch point; we are not interested in this precise point, only in its vicinity.

[4]Choosing $\delta\theta$ to be negative but of the same magnitude gives the same results.

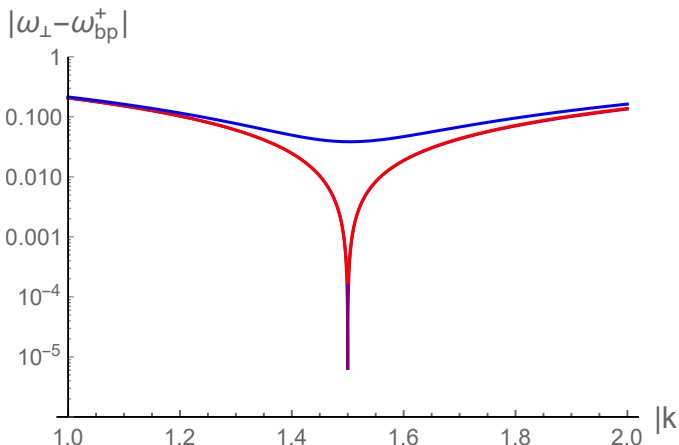

Figure 3: Distance in the complex $\omega$-plane between the hydrodynamic pole $\omega_\perp$ and the branch point $\omega_{bp}^+$ as $k$ varies along the ray $k_\theta = |k|e^{i\theta}$, for $\theta = \frac{\pi}{2} - 10^{-1}$ (blue), $\frac{\pi}{2} - 10^{-3}$ (red) and $\frac{\pi}{2} - 10^{-6}$ (purple). As $\frac{\pi}{2} - \theta \to 0$, the distance reaches a minimum at $|k| = \frac{3}{2}$.

where the parameter $\alpha$ is constrained to obey the transcendental equation

$$(\alpha^2 - 1)\left(2\pi n - i \log \frac{\alpha - 1}{\alpha + 1}\right) - 2i\alpha = 0. \tag{20}$$

The important point to keep in mind is that $\left(k = \pm\frac{3}{2}i, \omega = \frac{i}{2}\right)$ are still valid solutions in any non-principal sheet, since the replacement $L \to L_n$ leaves the conditions (15) invariant. In this sense, at $\left(k = \pm\frac{3}{2}i, \omega = \frac{i}{2}\right)$ the complex curve equation $P_\perp = 0$ goes from having an infinite number of solutions to having just one. It is natural to guess that, for each $n \neq 0$, there is going to be at least one $\omega_\perp^{(NH,n)}$ passing precisely through this point. Our numerical computations support this hypothesis. In figure 4, we plot the trajectories followed by a subset of the $\omega_\perp^{(NH,n)}$ poles as $k$ varies from 0 to $\frac{3}{2}i$ along the imaginary axis. We clearly see that, at $k = \frac{3}{2}i$, the poles degenerate. The relevant $\omega_\perp^{(NH,n)}$ have $\text{Re}(\alpha) > 0$ and $\text{Im}(\alpha) < 0$ ($> 0$) for $n < 0$ ($> 0$). An analogous situation takes place if we consider that $k$ ranges from 0 to $-\frac{3}{2}i$ instead; in this latter case, the $\omega_\perp^{(NH,n)}$ poles that become degenerate have the opposite value of $\text{Re}(\alpha)$.

To conclude this section, we discuss the behavior of both $\omega_\perp$ and the correlator (5) in the vicinity of $k = \frac{3}{2}i$. Let us start with the former object, and define

$$k = \frac{3i}{2} - i\delta, \quad \omega = \frac{i}{2} - i\delta - i\epsilon, \tag{21}$$

in such a way that $|\epsilon|$ measures the distance between $\omega_\perp$ and the branch point $\omega_{bp}^+$. A numerical analysis shows that, in the $\delta \to 0$ limit, $\epsilon$ behaves as

$$\epsilon \sim \frac{\delta}{-\log(\delta)}, \tag{22}$$

implying that $k = \frac{3}{2}i$ corresponds to a logarithmic branch point of $\omega_\perp$. We illustrate the behavior represented by equation (22) in figure 5 – where we consider that $\delta \in \mathbb{R}, \delta > 0$, in such a way that we approach $k = \frac{3}{2}i$ from below along the imaginary axis – and figure 6, where we provide the results for another two, different directions.

Regarding the correlator (5), it is natural to wonder the effect that the branch point-hydrodynamic pole collision we have found has on its residue at the pole. In particular, for

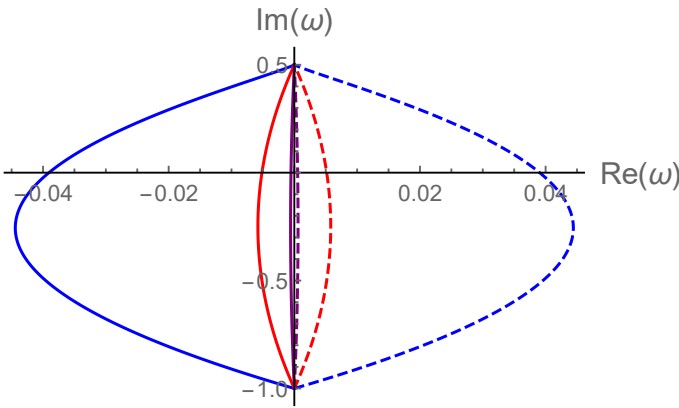

Figure 4: As $k$ goes from 0 to $\frac{3}{2}i$, the gapped poles in the non-principal sheets start at $\omega = -i$ (on their respective sheet) and degenerate at the branch point. We plot the trajectories followed by the poles with $|n| = 1$ (blue), $|n| = 10$ (red) and $|n| = 100$ (purple) with solid (dashed) curves corresponding to positive (negative) $n$.

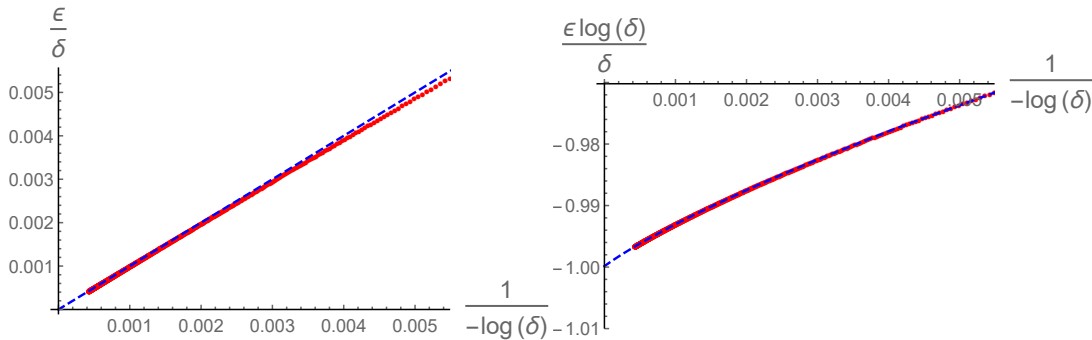

Figure 5: Left: numerically determined ratio $\frac{\epsilon}{\delta}$ as a function of $\frac{1}{-\log(\delta)}$ (red dots) together with the function $\frac{1}{-\log(\delta)}$ (dashed blue line). Right: numerically determined ratio $\frac{\epsilon \log(\delta)}{\delta}$ as a function of $\frac{1}{-\log(\delta)}$ (red dots), and corresponding interpolating function (solid blue). The values below $\frac{1}{-\log(\delta)} = 4.34294 \times 10^{-4}$ in the dashed blue curve have been obtained by an extrapolation. For $\delta = 0$, this extrapolated curve hits $-0.99985$, in very good agreement with equation (22).

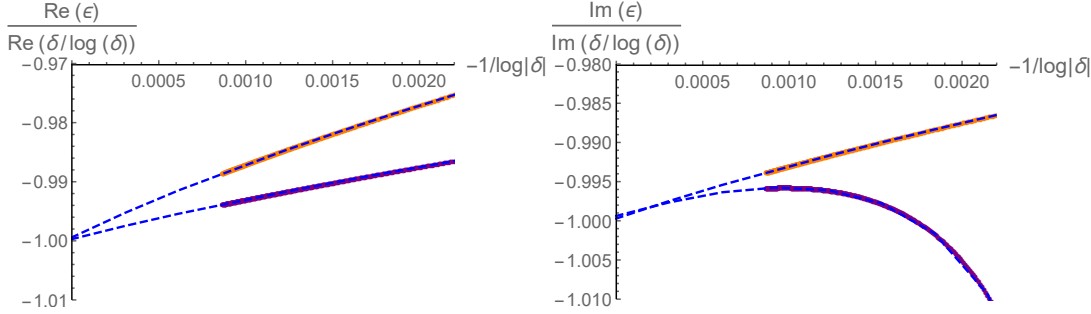

Figure 6: Check that the expression (22) holds when approaching $k = \frac{3}{2}i$ along the rays $k = \frac{3}{2}i + \xi e^{i\left(\frac{\pi}{2} - \frac{1}{100}\right)}$ (purple) and $k = \frac{3}{2}i + \xi$ (orange), with $\xi \in \mathbb{R}^+$. The dashed blue curves correspond to interpolating functions, whose extrapolation to $\delta = 0$ results in values compatible with $-1$. Left: check of the real part of (22). Right: check of the imaginary part.

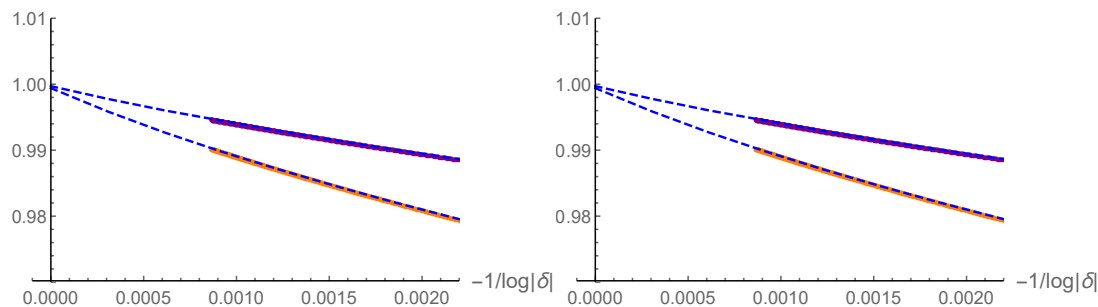

Figure 7: Ratios $\frac{\text{Re}(R_\perp^{01,01})}{\text{Re}(3i/4\,1/\log\delta)}$ (orange dots) and $\frac{\text{Im}(R_\perp^{01,01})}{\text{Im}(3i/4\,1/\log\delta)}$ (purple dots) when approaching $k = \frac{3}{2}i$ along the ray $k = \frac{3}{2}i + \xi e^{i\theta}$, with $\xi \in \mathbb{R}^+$. The dashed blue lines correspond to interpolating functions that, when extrapolated to $\delta = 0$, are compatible with one. Left: $\theta = \frac{\pi}{2} - \frac{1}{100}$. Right: $\theta = -\frac{\pi}{2} + \frac{1}{100}$.

the picture presented so far to be internally consistent, this residue would need to vanish right at the critical momentum where the collision takes place. As we illustrate in figure 7, this is precisely what happens. We considered the quantity

$$R_\perp^{01,01} = \text{Res}_{\omega=\omega_\perp(k)} \left( \frac{G_{R,\perp}^{01,01}(\omega,k)}{-(\mathcal{E}+\mathcal{P})} \right), \tag{23}$$

and explored its behavior as a function of $k$ when approaching $k = \frac{3}{2}i$ from different directions. Our observations are compatible with the functional form

$$R_\perp^{01,01} \sim \frac{3i}{4} \frac{1}{\log(\delta)}, \tag{24}$$

as $|\delta| \to 0$. The behavior of $\omega_\perp$ and $R_\perp^{01,01}$ around $k = -\frac{3}{2}i$ also follows (22) and (24) respectively, provided one replaces $k \to -k$ in the definitions of $\delta$ and $\epsilon$.

## 3 The sound channel

In the sound channel (6), the hydrodynamic mode frequencies $\omega_\parallel^\pm(k)$ are given by

$$P_\parallel(\omega,k) = 2k(k^2+3i\omega) + i(k^2+3\omega(i+\omega))L = 0, \tag{25}$$

where $L$ is defined as in equation (7). As before, $P_\parallel$ has two branch points located at $\omega_{bp}^\pm(k)$. The hydrodynamic mode frequencies $\omega_\parallel^\pm(k)$ behave as

$$\omega_\parallel^\pm(k) = \pm\frac{k}{\sqrt{3}} - i\frac{2}{15}k^2 + \dots. \tag{26}$$

The complete series expansion of $\omega_\parallel^\pm$ around $k=0$ has the form

$$\omega_\parallel^\pm(k) = \pm\frac{k}{\sqrt{3}} + \sum_{q=2}^\infty c_q^\pm k^q, \tag{27}$$

with $c_{2q+1}^- = -c_{2q+1}^+$ and $c_{2q}^- = c_{2q}^+$, due to the the fact that the hydrodynamic sound modes are symmetric with respect to the imaginary $\omega$-axis for real $k$, $\omega_\parallel^-(k) = -\omega_\parallel^+(k)^*$. In the following, we will focus on the $\omega_\parallel^-$ hydrodynamic mode.

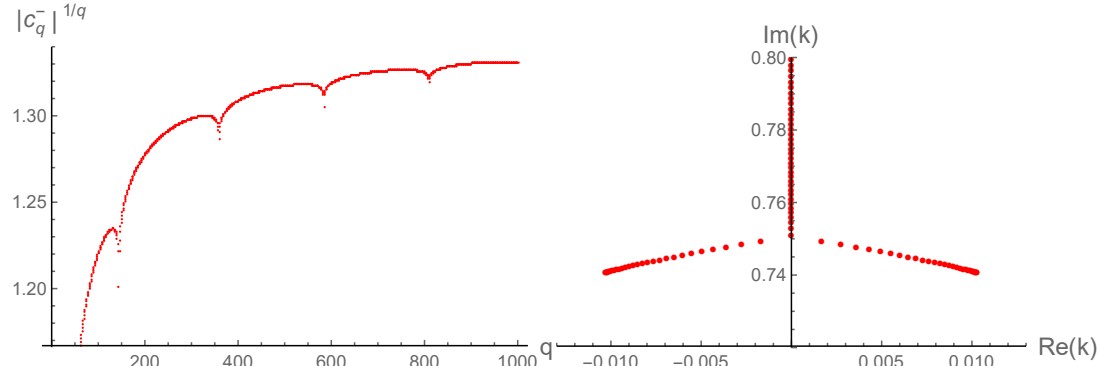

Figure 8: Left: root test applied to the $c_q^-$ coefficients of the series expansion of $\omega_\parallel^-$ around $k = 0$. The resulting sequence seems to saturate to a finite value as $q \to \infty$ in a non-monotonic fashion. Left: location in the complex $k$-plane of the poles of the symmetric Padé approximant (order 500) to the series expansion of $\omega_\parallel^-$ around $k = 0$ (truncated at order 1000). Three lines of pole condensation are clearly visible.

Plugging (27) into (25), series expanding around $k = 0$ and demanding that the resulting expression vanishes order-by-order allows us to determine the $c_q^-$ coefficients. We have carried out this procedure up to a maximum order $q = q_{max} = 10^3$. The result of applying the root test to the coefficient sequence can be found in figure 8 (left). We observe that $|c_q^-|^{\frac{1}{q}}$ saturates to a finite value in a non-monotonic fashion. In order to find the location of the singularities of $\omega_\parallel^-(k)$ in the complex $k$-plane, we first continue analytically the sum (27) – truncated to order $q_{max}$ – by means of a symmetric Padé approximant of order $\frac{q_{max}}{2}$, and then determine the locations of the poles of the resulting rational function. The three lines of pole condensation which are closest to $k = 0$ are depicted in figure 8 (right). One extends along the positive imaginary axis and emanates from the point

$$k_0 = 0.7513375i. \tag{28}$$

The other two are symmetric with respect to the imaginary axis, and start from the points

$$k_\pm = \pm 0.0102799 + 0.7409764i. \tag{29}$$

Since $|k_\pm| = 0.7410477 < |k_0|$, the symmetric points $k_\pm$ seem to be the ones setting the

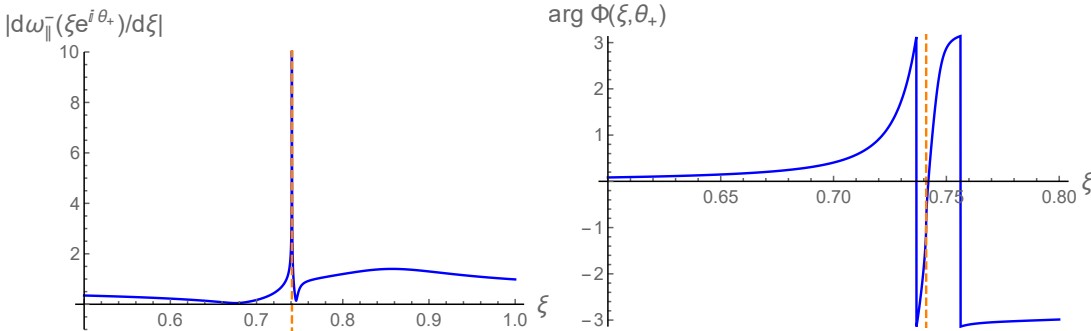

Figure 9: For the hydrodynamic mode frequency $\omega_\parallel^-$, computed along the ray $k = \xi e^{i\theta_+}$, norm of the first derivative with respect to $\xi$ (left plot) and argument of $L$ along the corresponding path in the complex $k$-plane (right plot). The point $\xi = |k_+|$ has been signalled by the dashed orange vertical line in both plots.

convergence radius of the series expansion of $\omega_\parallel^-(k)$ around $k = 0$.

In order to understand what these symmetric poles correspond to, we transform the original complex curve equation $P_\parallel(\omega_\parallel(k), k) = 0$ into an ODE for $\omega_\parallel(k)$, just as we described in the previous section for the shear channel. This ODE reads

$$C(\omega_\parallel(k), k)\omega_\parallel'(k) - D(\omega_\parallel(k), k) = 0,\tag{30}$$

$$C(\omega, k) = ik^2 + 6k^3\omega + 3ik\omega^2 - 6k\omega^3,\tag{31}$$

$$D(\omega, k) = k^4 + 5ik^2\omega + 8k^2\omega^2 - 9i\omega^3 - 9\omega^4.\tag{32}$$

Solving this ODE along the ray $k = \xi e^{i\theta_+}$, where $\theta_+ = \arg k_+$ and $\xi \in \mathbb{R}^+$, we find the results displayed in figure 9. In the left plot, we observe that at, $\xi = |k_+|$,

$$\left|\frac{d}{d\xi}\omega_\parallel^-(\xi e^{i\theta_+})\right|$$

peaks. Hence, $k_+$ is close to a point in which $\omega_\parallel^-(k)$ has a divergent first derivative. In the right plot, we show the phase of the argument of $L$, $\Phi = \frac{\omega - k + i}{\omega + k + i}$, evaluated along our path. We see that the point $k = k_+$ is reached after $\omega_\parallel^-(k)$ has crossed the branch cut and entered into the $n = 1$ sheet.[5]

In order to find the critical momentum at which $\omega_\parallel^{-\prime}(k)$ diverges, we take advantage of the fact that $\omega_\parallel'(k)$ obeys

$$\partial_\omega P_\parallel(\omega_\parallel(k), k)\omega_\parallel'(k) + \partial_k P_\parallel(\omega_\parallel(k), k) = 0,\tag{33}$$

and, as a consequence, points for which

$$P_\parallel(\omega, k) = \partial_\omega P_\parallel(\omega, k) = 0, \quad 0 < \left|\partial_k P_\parallel(\omega, k)\right| < \infty,\tag{34}$$

have a divergent $\omega_\parallel^{\pm\prime}(k)$. A crucial observation is that, since the point we are after lies on the $n = 1$ sheet, we need to first analytically continue $L \to L_{n=1}$ as described in the previous section. In the end, a numerical computation reveals that the momentum $k_{c,n=1}$ at which conditions (34) hold is given by

$$k_{c,n=1} = 0.0102873 + 0.7409673i\,.\tag{35}$$

The distance between $k_{c,n=1}$ and $k_+$ is $1.17 \times 10^{-5}$. This confirms that the right point of pole accumulation observed in the Padé approximant is actually associated with a point in which $\omega_\parallel^{-\prime}(k)$ diverges. Analogous arguments can be employed to demonstrate that $k_-$ is associated to a divergent $\omega_\parallel^{-\prime}(k)$ in the $n = -1$ sheet.

It is natural to wonder whether points in which $\omega_\parallel^{-\prime}(k)$ diverges are restricted to the $n = \pm 1$ sheets. The answer is negative: for every nonzero $n \in \mathbb{Z}$, there exists a point $k_{c,n}$ of this kind, which can be found by solving (34) on the $n$-th sheet. Since, numerically, we find that $k_{c,-|n|} = -k_{c,|n|}^*$, we only discuss the $n > 0$ case in the following. The behavior of $k_{c,n}$ is illustrated in figure 10. We find that $\text{Re}(k_{c,n})$ decreases monotonically with $n$, approaching zero as $n \to \infty$. On the other hand, both $\text{Im}(k_{c,n})$ and $|k_{c,n}|$ increase monotonically, tending to $\frac{3}{4}$ in the $n \to \infty$ limit. Hence, the singularities of $\omega_\parallel^-(k)$ which are closest to the origin correspond to $k_{c,\pm 1}$: these are the singularities that set the convergence radius of the series expansion of $\omega_\parallel^-$ around $k = 0$.

---

[5]This observation follows from the fact that $\arg \Phi$ flips from $+\pi$ to $-\pi$.

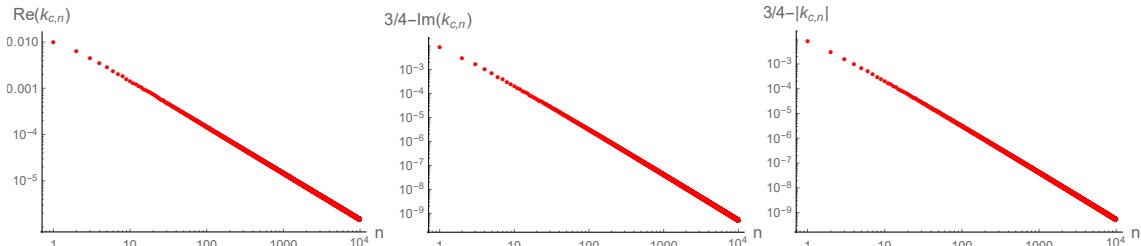

Figure 10: Numerically determined values of the points $k_{c,n}$ (for $n > 0$) where the relations (34) are obeyed in the upper-half of the complex $k$-plane close to the imaginary axis. At these points, $\omega_{\parallel}^{-\prime}(k)$ diverges. We plot their real part (left plot), their imaginary part (middle plot) and their norm (right plot).

An immediate consequence of the results presented above is that the point $k = \frac{3}{4}i$ acts as an accumulation point of two infinite sequences of branch points, one coming in from the right, associated with positive $n$, and another coming in from the left, associated with negative $n$.[6] Furthermore, the point $k = \frac{3}{4}i$ seems to correspond to the first purely imaginary pole of our original Padé approximant (cf. equation (28)). As we argue in appendix A, despite being associated to the starting point of a line of pole condensation of a Padé approximant, $k = \frac{3}{4}i$ does not correspond to a branch point of $\omega_{\parallel}^{-}$, but rather to an essential singularity.[7]

To conclude this section, we will show that the critical momenta $k_{c,n}$ can be understood as arising from a pole collision. As for the shear channel case, the solutions of the complex curve equation $P_{\parallel} = 0$ on the non-principal sheets play a prominent role in our analysis. In the case at hand, the solutions of interest are the non-principal gapless poles,[8] whose series expansion around $k = 0$ is given by

$$\omega_{\parallel}^{(H,n)}(k) = \frac{i}{3}k^2 - \frac{i}{9}k^4 - \frac{i}{9\pi n}k^5 + \frac{2i}{27}k^6 + \frac{4i}{27n\pi}k^7 + \dots \tag{36}$$

Our main claim is that the critical momenta $k_{c,n}$ correspond to branch point singularities at which the hydrodynamic $\omega_{\parallel}^{-}$ mode collides with $\omega_{\parallel}^{(H,n)}$ on the $n$-th sheet of $P_{\parallel}$.

In order to illustrate this statement, let us consider the behavior of $\omega_{\parallel}^{(H,1)}$ and $\omega_{\parallel}^{-}$ in the complex $\omega$-plane as we vary $k$ along the ray $k = |k|e^{i\theta}$. In figure 11 we plot the trajectories of $\omega_{\parallel}^{-}$ (solid) and $\omega_{\parallel}^{(H,1)}$ (dashed) for $\theta - \arg k_{c,1} = -10^{-2}$ (brown), $-10^{-3}$ (red), $-10^{-4}$ (blue) and $-10^{-5}$ (purple). As $\theta - \arg k_{c,1} \to 0$, both $\omega_{\parallel}^{-}$ and $\omega_{\parallel}^{(H,1)}$ approach the point

$$\omega_{\parallel}^{-}(k_{c,1}) = 0.0142827 - 0.2563247i\,, \tag{37}$$

more closely. This point, which has been determined independently by solving the conditions (34) for $n = 1$, corresponds to the green star in the figure. The behavior we observe is precisely the one to expect if there is indeed a mode collision at $\omega = \omega_{\parallel}^{-}(k_{c,1})$.

Regarding the $\omega_{\parallel}^{+}(k)$ hydrodynamic mode, an analysis analogous to the one presented so far shows that the complex singularities closest to the origin correspond to two branch points located at $k = k_{c,\pm 1}^{*}$, which can again be interpreted as collisions between $\omega_{\parallel}^{+}(k)$ and $\omega_{\parallel}^{(H,\pm 1)}$ on non-principal sheets.

---

[6]A direct numerical analysis shows that $\arg\left(\frac{3}{4}i - k_{c,n}\right) \to \pi$ as $n \to \infty$, i.e., the curves along which the branch points condense are parallel to the real $k$-axis in the $n \to \infty$ limit

[7]Some authors require an essential singularity to be isolated, but we do not.

[8]There are also non-principal gapped poles, with series expansion $\omega_{\parallel}^{(NH,n)} = -i + \alpha k + \frac{1}{3}i(\alpha^2 - 1)k^2 + \dots$, where $\alpha$ obeys a transcendental equation. For $n = 0$, this equation has no solutions; for $n \neq 0$, we find purely imaginary solutions – with positive imaginary part for $n > 0$ – that the change $n \to -n$ conjugates.

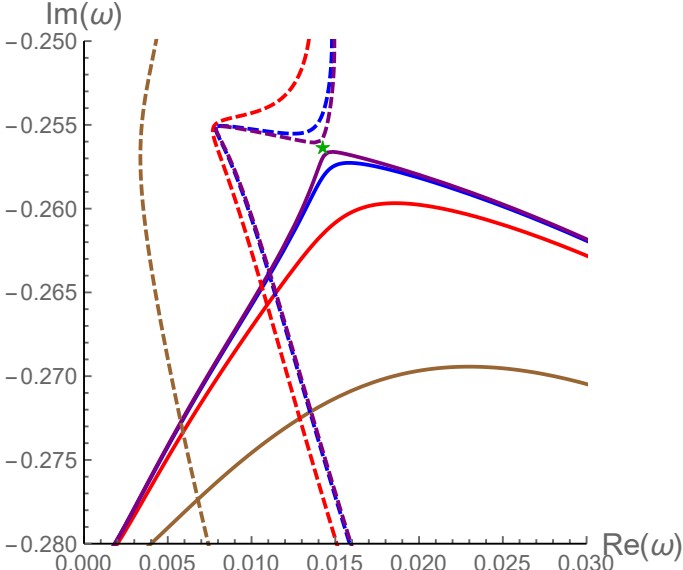

**Figure 11:** Trajectories followed by $\omega_{\parallel}^{-}$ (solid curves) and $\omega^{(H,n=1)}$ (dashed curves) in the complex $\omega$-plane, as $k$ varies along the ray $k = |k|e^{i\theta}$ for $\theta - \arg k_{c,1} = -10^{-2}$ (brown), $-10^{-3}$ (red), $-10^{-4}$ (blue) and $-10^{-5}$ (purple). As $\theta - \arg k_{c,1}$ decreases, the corresponding trajectories become closer to each other, until colliding at the point $\omega = 0.0142827 - 0.2563247i$ (green star). This point has been determined independently by solving equations (34) for $n = 1$.

Taking stock, the main conclusion of the analysis presented in this section is that the convergence radius of the series expansion of $\omega_{\parallel}^{\pm}(k)$ around $k = 0$ is given by

$$|k_{\parallel}^{*}| = |k_{c,1}| = 0.7410387, \tag{38}$$

and set by mode collisions between $\omega_{\parallel}^{\pm}(k)$ and $\omega_{\parallel}^{(H,\pm1)}$ on non-principal sheets.

# 4 A prescription for finding the radius of convergence

In the preceding sections our analysis of RTA kinetic theory has revealed novel obstructions to the convergence of the hydrodynamic series. In light of this, it is worth revisiting the basics surrounding convergence of the function $\omega(k)$ expanded as a series in small $k$. The radius of convergence of $\omega(k)$ expanded around any value of $k$ (which we will mostly take to be $k = 0$, relevant to the hydrodynamic expansion) is determined by the closest singularity of $\omega(k)$ to that point in the complex $k$-plane.[9] Often, $\omega(k)$ is implicitly defined by a complex curve given by

$$P(\omega, k) = 0, \tag{39}$$

which may correspond to the determinant of a fluctuation mode matrix or, with care, the inverse of an appropriate Green's function or simply its denominator. These elementary observations have previously been used to determine the radius of convergence of the hydrodynamic gradient expansion for holographic theories [8–10, 13, 14, 16] as well as MIS theory [12]. In

---

[9]If $\omega(k)$ is defined on a multi-sheeted Riemann surface, one must also ensure the singularity is on the correct sheet of $\omega$.

the preceding sections of this paper we have employed this procedure to compute this radius of convergence for RTA kinetic theory.

A natural question is given $P(\omega, k)$, is there a shortcut to determining the singularities of $\omega(k)$? In this section we examine this question in the light of the Implicit Function Theorem. In doing so, we arrive at a prescription for a set of points that includes all singularities of $\omega(k)$. For previous related work on a similar question we direct the reader to [9, 10], although we note our prescription differs in some key mathematical aspects.

Let us first recall the Implicit Function Theorem (taken from [36])

> **Implicit Function Theorem.** Let $P(\omega, k)$ be analytic in $\omega, k$ near $\omega = k = 0$. Assuming $P(0, 0) = 0$ and $\partial_\omega P(0, 0) \neq 0$ then there exists a unique function $\omega(k)$ analytic in some neighbourhood $|k| < k_*$ of $k = 0$ such that $\omega(0) = 0$ and $P(\omega(k), k) = 0$.

Conversely, the following set of points $\omega, k \in \mathbb{C}$,

$$\{(\omega, k) \,|\, P = \partial_\omega P = 0\} \cup \{(\omega, k) \,|\, P = 0 \wedge (P \text{ is not analytic})\}, \tag{40}$$

includes the locations of all singularities of $\omega(k)$, since these are the only possible points of $P = 0$ where the function $\omega(k)$ may itself fail to be analytic by the Implicit Function Theorem.[10] Crucially however, note that the set (40) can include points where there are no singularities of $\omega(k)$. Thus, given the set (40), to determine the radius of convergence each point should be further examined in order to find out which, if any, is the closest singularity to the point around which one is expanding (which is $\omega = k = 0$ for the hydrodynamic expansion).

In the remainder of this section we illustrate these observations with a set of examples, beginning with the relatively simple case where $P$ is a polynomial in section 4.1, before discussing our main results on RTA kinetic theory and how they fit into this picture, for which $P$ is not a polynomial, in section 4.2.

## 4.1 Polynomial $P$

Here we collect some known results from [36]. Suppose $P$ can be written as a polynomial in $\omega$ and $k$ as follows

$$P(\omega, k) = \sum_{i=0}^{N} c_i(k) \omega^i. \tag{41}$$

In this case $\omega(k)$ is said to be an algebraic function. The candidate singularities (40) thus either correspond to those points where $P = \partial_\omega P = 0$, in which case multiple branches degenerate, or where $P$ is non-analytic. Since $P$ is polynomial, if $k$ is finite, this latter case only happens when $\omega = \infty$. This occurs only when the coefficient of the highest order term in $\omega$ vanishes, $c_N(k) = 0$. In this case, the number of roots of the resulting polynomial in $\omega$ is reduced.

We can also illustrate a case where a point in the set (40) does not correspond to a singularity of $\omega(k)$. This occurs whenever two branches of $\omega(k)$ happen to take on the same value at some $k \in \mathbb{C}$, with each branch remaining analytic around this point. The canonical example is the Lemniscate of Bernoulli [37] where, at the origin of the 'figure-of-eight'-shaped curve obeying $P = \partial_\omega P = 0$, two branches cross but remain analytic there.[11]

The type of singularities that algebraic functions can have is constrained by the Newton-Puiseux theorem. If $P$ has $r$ degenerate roots at a point, the theorem says that this singularity is a branch point of order $r$. Furthermore, around this point, $\omega(k)$ can be expanded as a

---

[10]In [9, 10] $P$ was labelled a 'spectral curve' and focus placed on points for which $P = \partial_\omega P = 0$, labelled 'critical points'.

[11]A similar phenomenon occurs in holography, where branches of the scalar quasinormal mode spectrum for the BTZ black hole [38, 39], $\omega_n(k)$, cross at finite $k \in \mathbb{C}$, but with no corresponding singularity of $\omega_n(k)$.

convergent series in powers of $k^{1/r}$.[12] In the next section we show that this is no longer the case for non-polynomial $P$.

## 4.2 Non-polynomial $P$ and case studies

We now turn to the instances where $P(\omega, k)$ is not a polynomial. This may be the case when it arises as an all-orders gradient expansion, or includes functions like the logarithm as in RTA kinetic theory above. Then $P$ can fail to be analytic in more ways than a polynomial is allowed to. For example, there can be points where $P$ does not converge (as it happens in the case of the gradient expansion), or where some of its pieces are non-analytic. Again, we stress that these conditions do not imply that there is necessarily a singularity there, only that they are permitted.

### 4.2.1 Single mode

To examine this case in the simplest way possible, consider a theory with a single mode,

$$P(\omega, k) = \omega - \omega_1(k). \tag{42}$$

We give a physical example below. Regarding the set (40), there are no points for (42) where $\partial_\omega P = 0$. However, the set of potential singularities (40) need not be empty, since $\omega_1(k)$ itself can contain singularities. These are then inherited by $\omega(k)$.

This scenario occurs in the following physical example: relativistic hydrodynamics, defined perturbatively in gradients. Without loss of generality, the gradient-expanded constitutive relations can be put in the Landau frame, and when evaluated on the hydrodynamic fluctuations can be reorganized using the equations of motion such that they contain no time derivatives [12, 40]. The equations of motion for a shear channel fluctuation therefore contain one and only one time derivative, which acts on the ideal part of the current. The resulting $P(\omega, k)$ is necessarily of the form (42) with $\omega_1(k)$ expressed as an infinite series in $k$. As an illustrative concrete example, the hydrodynamic limit of MIS theory in the shear channel has a $P$ of the form (42) where [12],

$$\omega_1(k) = -i \sum_{n=0}^{\infty} \mathcal{C}_n D^{n+1} \tau^n k^{2n+2}, \tag{43}$$

where $\mathcal{C}_n$ are the Catalan numbers, and $D, \tau$ are respectively the diffusion constant and relaxation time. Note that the series (43) has a finite radius of convergence, and correspondingly summing (43) gives the exact expression for $\omega_1(k)$ which contains a branch point singularity.

### 4.2.2 RTA kinetic theory: shear channel

The case studied in this paper, RTA kinetic theory, offers several new interesting elements. These arise from the multi-sheeted structure of the logarithmic function which enters into $P$. We find that the singularities of $\omega(k)$ can be much richer and moreover can no longer be described by Puiseux series.

The singularity of $\omega_\perp(k)$ which is closest to the origin and that sets the radius of convergence occurs when the hydrodynamic pole collides with the logarithmic branch point. In fact, an infinite number of gapped poles located on the other non-principal sheets also collide with the branch point. Referring to the set of points (40), this point does not satisfy the condition $\partial_\omega P = 0$, rather, it is a point where $P$ is non-analytic. This occurs at finite $\omega$ and $k$, which is impossible in the polynomial case.

---

[12]If the singularity is located at $\omega = \infty$, this holds for $\omega^{-1}$ instead.

One may think of this singularity as an infinite-dimensional generalization of what happens when the highest order term vanishes for a polynomial. Indeed, at this point, the coefficient of the logarithm in $P$ vanishes, reducing the number of branches from an infinite number to a finite number. On the other hand, around this point there are an infinite number of branches in $\omega_\perp(k)$; if one expresses $\omega_\perp(k)$ as a series around this point, it does not take the form of a Puiseux series, since a Puiseux series can only describe a finite number of branches. Instead of fractional powers, the numerical evidence at our disposal indicates that the series includes logarithms (see equation (22)).

### 4.2.3 RTA kinetic theory: sound channel

In the sound channel, the multivaluedness of the logarithm also plays an interesting role. Surprisingly, each sheet of the analytically continued Green's function contains an additional gapless pole.

It is unclear whether these additional poles have any significant role to play in terms of physical excitations of the system; nevertheless, their mathematical role in our current analysis is clear. They give rise to singularities in $\omega_\parallel(k)$ at the values of $k$ where they collide with the physical hydrodynamic pole. This happens for each of the non-principal poles at a sequence of $k$ accumulating at an essential singularity of $\omega_\parallel(k)$. The radius of convergence of $\omega_\parallel(k)$ around $k = 0$ is given by singularities coming from the collision with $\omega_\parallel^{(H, n=\pm 1)}$. At this point, the analytic continuation of $P$ has a degeneracy as diagnosed by the $P = \partial_\omega P = 0$ condition in (40).

The essential singularity at the accumulation point in $\omega_\parallel(k)$ is a new feature not possible for algebraic curves. Around this point, a Puiseux series does not capture the behaviour of $\omega_\parallel(k)$. In appendix A, we show that the expansion includes exponentially small contributions, taking the form of a transseries.

## 5  Summary

The physics of nonequilibrium systems has benefited enormously from studies of model systems in which the approach to equilibrium and the emergence of hydrodynamic behaviour could be investigated. Until the advent of holography, the most prominent approach was to use kinetic theory, whose applicability rests upon the notion of well-defined particles (or quasi-particles) and the weak-coupling regime. In contrast, the AdS/CFT correspondence is used at infinitely strong coupling.

Microscopic models formulated in the language of holography or kinetic theory both lead to hydrodynamic behaviour close to equilibrium in a sense which can be made precise at the linearized level: the system has modes whose frequencies vanish at long wavelength. There is however a significant qualitative difference in the remaining features of the analytic structure of retarded correlators. While the singularities of strongly coupled theories as well as MIS-type models take the form of isolated poles, the retarded Green's function of kinetic theory in the relaxation time approximation has both poles and branch points. This implies that the natural object to deal with is the analytically continued Green's function, which can be viewed as defined on a multi-sheeted Riemann surface.

The analytically continued Green's function in RTA kinetic theory has an infinite number of poles. In the shear channel, there is a single hydrodynamic pole on the physical sheet, and an infinite number of nonhydrodynamic poles on the non-principal ones. The radius of convergence of the hydrodynamic series is set by a collision of the hydrodynamic pole and a branch point. In the sound channel, the radius of convergence of the hydrodynamic series is

set by the collision of the hydrodynamic pole with a gapless pole on a non-principal sheet. Note that in order to understand the emergence of the finite radius of convergence in terms of singularity collisions, it is essential to continue beyond the principal branch of the logarithm appearing in the original Green's function.

It is worth pointing out that the structure of gapped poles in kinetic theory found here is consistent with studies of boost-invariant flow in RTA kinetic theory. Calculations of the late proper time expansion in $\mathcal{N} = 4$ SYM reveal that its large-order behaviour contains detailed information about the rich nonhydrodynamic spectrum of this theory, which is completely consistent with what is known from linear response [41, 42]. Analogous calculations in MIS-type models [43, 44] show a similar consistency. Calculations of the late proper time expansion in RTA kinetic theory [45] point to the conclusion that the nonhydrodynamic sector consists of an infinite number of nonhydrodynamic modes whose frequencies coincide at vanishing momentum [46]. The analysis of analytically continued retarded correlation functions described here reveals such a set of gapped poles. Whether these objects can be mapped to each other remains an open problem.

More generally, our analysis has not addressed the question of whether the poles of the analytically continued Green's function in the non-principal sheets are endowed with any physical significance. While we don't have an answer to this question, we would like to point out that similar situations are not unprecedented. For instance, in the context of hadronic scattering, resonances appear as poles of the scattering amplitude in non-principal sheets [47]. Perhaps closer to the present context, poles in non-principal sheets are also relevant for the phenomenon of Landau damping in scalar quantum electrodynamics [48].

Whenever the dispersion relations are defined by a complex curve $P(\omega, k) = 0$, the Implicit Function Theorem implies that singularities are allowed (but not necessary) only at points where either $P$ is non-analytic or $\partial_\omega P = 0$. Both conditions must be taken into account. When $P$ is polynomial, the latter condition plays an important role since it can indicate a branch point singularity (but not always), and the former condition can indicate poles. In RTA kinetic theory, we have found that the former condition plays a significant role. In theories where $P$ is a polynomial, the behavior can be described by a Puiseux series with a non-zero radius of convergence. In theories where $P$ is not a polynomial, such as RTA kinetic theory, the singularity can be of other types and it is not known if the series expansion around these singularities is convergent or not. In holography, while the singularities of $\omega(k)$ which set the radius of convergence are known to be square-root type branch points in the examples studied to date,[13] the analytic structure of $P(\omega, k)$ in general remains an open question.

# Acknowledgements

We would like to thank R. A. Davison, B. Goutéraux and C. Pantelidou for valuable discussions. We would also like to thank the organisers and participants of the online seminar series HoloTube [49] for their hospitality. The Gravity, Quantum Fields and Information group at the Max Planck Institute for Gravitational Physics (Albert Einstein Institute) is supported by the Alexander von Humboldt Foundation and the Federal Ministry for Education and Research through the Sofja Kovalevskaja Award. AS and MS are supported by the Polish National Science Centre grant 2018/29/B/ST2/02457. BW is supported by a Royal Society University Research Fellowship.

---

[13]Potentially quartic roots are seen in special cases [14].

# A  The points $k = \pm\frac{3}{4}i$ in the sound channel

In this appendix we address the behavior of $\omega_\parallel^\pm(k)$ along the imaginary $k$-axis. The analysis presented here reveals the existence of essential singularities in the sound channel dispersion relations, although we would like to emphasize that these essential singularities occur outside the radius of convergence of the small-$k$ expansion of $\omega_\parallel^\pm(k)$.

We study $\omega_\parallel^-(k)$ first, and start by solving the equation of motion (30) around $k = \frac{3}{4}i$ in a series expansion, i.e., we define $k = i\left(\frac{3}{4} - \delta\right)$, $\delta \in \mathbb{R}$, $\delta > 0$ and consider the ansatz

$$\omega_\parallel^-(k) = iw_0 + i\sum_{q=1}^\infty w_q \delta^q. \tag{44}$$

At zeroth order, this results is four roots –with one degenerate root-, $w_0 = -\frac{1}{4}, \frac{1}{4}$ and $\frac{3}{4}$, of which the first one is singled out by a direct comparison with the numerical solution. The remaining expansion coefficients can be determined recursively, with the result that $w_1 = -1$ and the rest vanish: the behavior of $\omega_\parallel^-(k)$ as $k \to \frac{3}{4}i$ from below along the imaginary axis cannot be reproduced by a power series ansatz. To see this, let us linearize around the power series solution we just found. We consider

$$\omega_\parallel^-(k) = -\frac{1}{4}i - i\delta + \sum_{q=1}^\infty \chi_q(\delta)\eta^q, \tag{45}$$

where $\eta$ is a fictitious parameter, and solve (30) recursively in a $\eta \to 0$ expansion. In the end, we find that

$$\chi_1(\delta) = \alpha e^{-\frac{3}{32\delta}} e^{\frac{\delta}{2}} (3 - 4\delta), \tag{46}$$

$$\chi_2(\delta) = i\alpha^2 e^{-\frac{3}{16\delta}} e^\delta (3 - 4\delta) \frac{9 - 108\delta - 656\delta^2 + 192\delta^3}{64\delta^2}, \tag{47}$$

etc., with the overall conclusion that $\chi_q(\delta) = O\left(e^{-\frac{3}{32\delta}q}\right)$. Hence, there is an essential singularity at $\delta = 0$. Said otherwise, the distance between $\omega_\parallel^-(k)$ and the branch point at $\omega_{bp}^+(k) = k - i = -i\frac{1}{4} - i\delta$ is nonperturbatively small in $\delta$ as $\delta \to 0$. In the light of these results, we can set $\eta = 1$ in (45) and view the result as a transseries expansion in $\delta$.

The only point that this analysis cannot address is the value of the coefficient $\alpha$. To fix it, we need to plug the transseries expansion (45) back into $P_\parallel(\omega, k)$, expand around $\delta = 0$, and demand that we have a solution. The end result of this procedure is that

$$\alpha = \frac{i}{2}e^{-\frac{9}{4}}. \tag{48}$$

As a final consistency check, we compare our transseries expansion truncated to order $q = q_{max}$ with the $\omega_\parallel^-$ we have determined numerically. It is convenient to plot the ratio

$$r_{q_{max}}^- = \frac{\omega_\parallel^-(k) - \omega_{bp}^+(k)}{\sum_{q=1}^{q_{max}} \chi_q\left(\delta = \frac{3}{4} + ik\right)}. \tag{49}$$

Examples for $q_{max} = 1$, 4 and 8 can be found in figure 12 (right). We see that $r_{q_{max}}^- \to 1$ as $k \to \frac{3}{4}i$; furthermore, as $q_{max}$ increases, the agreement away from $k = \frac{3}{4}i$ improves significantly. This confirms that our analysis is correct. The distance between $\omega_\parallel^-$ and the branch point $\omega_{bp}^+$ can be found in the left plot of figure 12.

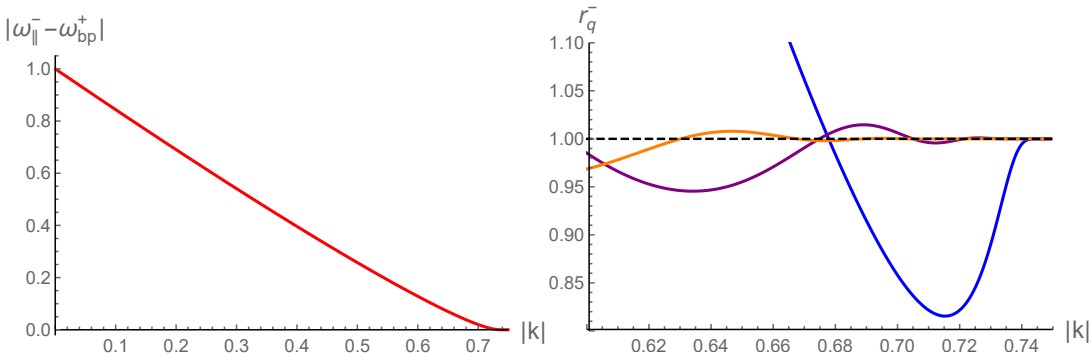

Figure 12: Left: distance between the hydrodynamic mode frequency $\omega_{\parallel}^{-}$ and the branch point $\omega_{bp}^{+}$, as $k$ goes from 0 to $\frac{3}{4}i$ along the imaginary axis. This distance decreases monotonically and vanishes as $k \to \frac{3}{4}i$. Right: comparison between the distance $|\omega_{\parallel}^{-} - \omega_{bp}^{+}|$ and the analytic prediction (45) for $k \to \frac{3}{4}i$ along the imaginary $k$-axis from below, as quantified by the ratio $r_q^{-}$ defined in equation (49). We have considered $q = 1$ (blue), $q = 4$ (purple) and $q = 8$ (orange). As $q$ increases, the agreement gets progressively better, as seen by the progressively closer approach of the different curves to the value $r_q^{-} = 1$ (dashed black line).

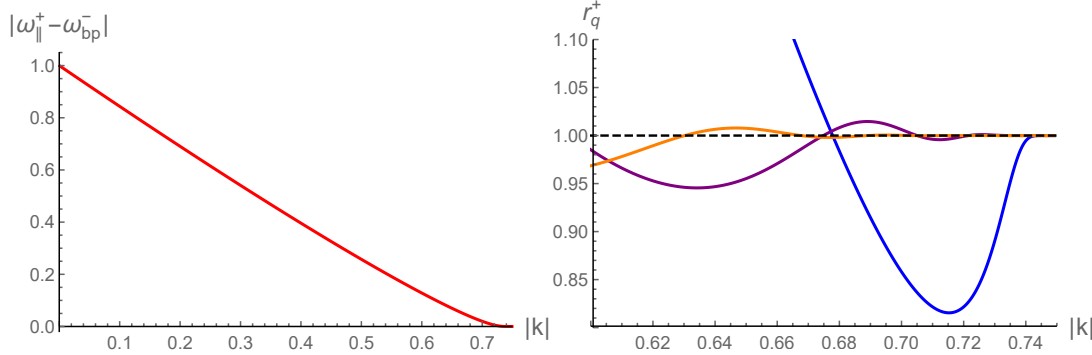

Figure 13: Same as figure 12, but now for the hydrodynamic mode frequency $\omega_{\parallel}^{+}$ and the branch point $\omega_{bp}^{-}$. The relevant modifications that need to be performed to obtain this figure are described in the text.

An analysis analogous to the one presented above can be carried out for $\omega_{\parallel}^{+}$. In this case, we find that this mode collides with the $\omega_{bp}^{-}$ branch point at $k = -\frac{3}{4}i$. Defining $\delta$ by the relation $k = -\frac{3}{4}i + \delta i$, $\omega_{\parallel}^{+}$ can also be represented as a transseries in $\delta$ as $\delta \to 0$,

$$\omega_{\parallel}^{+} = -\frac{i}{4} - i\delta + \sum_{q=1}^{\infty} \chi_q(\delta), \tag{50}$$

with $\chi_q(\delta)$ given by the same expressions as above. We check the relation (50) –truncated at orders $q_{max} = 1$, 4 and 8– in figure 13 (right), where we plot the ratio

$$r_{q_{max}}^{+} = \frac{\omega_{\parallel}^{+}(k) - \omega_{bp}^{-}(k)}{\sum_{q=1}^{q_{max}} \chi_q\left(\delta = \frac{3}{4} - ik\right)}. \tag{51}$$

To summarize, in this appendix we have demonstrated that:

- $\omega_\parallel^\pm(k)$ present essential singularities at $k = \mp\frac{3}{4}i$.

- These essential singularities appear when the $\omega_\parallel^\pm$ pole collides with the $\omega_{bp}^\mp$ logarithmic branch point.

As discussed in the main body of the text, this phenomenon has no counterpart for algebraic functions.

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
