# Peer review of "Convergence of hydrodynamic modes: insights from kinetic theory and holography"

_SciPost Physics, doi:SciPost Phys. 10, 123 (2021)_

## Round 1 · Referee Report · Anonymous · 2021-3-11

Strengths
1- very well written, clear structure
2- results are new and timely
3- discussion of results is sufficiently detailed, but always to the point
Report
The authors consider hydrodynamic shear and sound dispersion relations $\omega = \omega(k)$ and study the radius of convergence of the expansion around $k=0$ using kinetic theory (weak coupling). In the relaxation time approximation, the relevant retarded Green's functions are known explicitly and have poles as well as branch points. The paper provides evidence that the radius of convergence of the hydrodynamic expansion is given by the location where hydrodynamic poles collide with the logarithmic branch point or a pole on the second sheet of the retarded Green's function. The evidence is mostly numerical. The numerical results are convincing (the authors provide detailed consistency checks) and appear to be correct.
The paper is very well written and structured, explanations are clear, detailed and thorough. The results are new and timely, making contact with numerous recent developments on the convergence of the hydrodynamic expansion in holography (and MIS theory) and offering novel viewpoints.
Regarding SciPost's acceptance criteria: while the content of the paper may not be groundbreaking, I think it does "provide a novel and synergetic link between different research areas" by establishing interesting connections and contrasts between strongly vs. weakly coupled hydrodynamics, while also offering some conjectures and open questions for future research. All general acceptance criteria are met as far as I can tell. I would recommend the paper for publication as is.

---

## Editorial Decision

published